# Melatonin Positively Regulates Both Dark- and Age-Induced Leaf Senescence by Reducing ROS Accumulation and Modulating Abscisic Acid and Auxin Biosynthesis in Cucumber Plants

**DOI:** 10.3390/ijms23073576

**Published:** 2022-03-25

**Authors:** Tongtong Jing, Kun Liu, Yanan Wang, Xizhen Ai, Huangai Bi

**Affiliations:** State Key Laboratory of Crop Biology, Key Laboratory of Crop Biology and Genetic Improvement of Horticultural Crops in Huanghuai Region, College of Horticulture Science and Engineering, Shandong Agricultural University, Tai’an 271018, China; J898872118@163.com (T.J.); ulking1223@163.com (K.L.); 18864805562@163.com (Y.W.); axz@sdau.edu.cn (X.A.)

**Keywords:** melatonin, leaf senescence, antioxidant system, auxin, abscisic acid, cucumber

## Abstract

Melatonin (MT), as a signaling molecule, plays a vital role in regulating leaf senescence in plants. This study aimed to verify the antioxidant roles of MT in delaying dark- or age-induced leaf senescence of cucumber plants. The results showed that endogenous MT responds to darkness and overexpression of *CsASMT*, the key gene of MT synthesis, and delays leaf senescence stimulated by darkness, as manifested by significantly lower malonaldehyde (MDA) and reactive oxygen species (ROS) contents as well as higher activities and gene expression of antioxidant enzymes compared to the control. Moreover, MT suppressed both age- or dark-induced leaf senescence of cucumber, as evidenced by a decrease in senescence-related gene *SAG20* and cell-death-related gene *PDCD* expression and ROS content and an increase in antioxidant capacity and chlorophyll biosynthesis compared with the H_2_O-treated seedlings. Meanwhile, the suppression of age-induced leaf senescence by melatonin was also reflected by the reduction in abscisic acid (ABA) biosynthesis and signaling pathways as well as the promotion of auxin (IAA) biosynthesis and signaling pathways in cucumber plants in the solar greenhouse. Combining the results of the two separate experiments, we demonstrated that MT acts as a powerful antioxidant to alleviate leaf senescence by activating the antioxidant system and IAA synthesis and signaling while inhibiting ABA synthesis and signaling in cucumber plants.

## 1. Introduction

Cucumber (*Cucumis sativus* L.) is an important vegetable crop that is cultivated worldwide. However, leaf senescence induced by age or abiotic stress leads to an obvious decrease in photosynthetic efficiency, which further results in a decline in production because of the shortened harvest time [1,2,3], particularly during the cultivation pattern of one crop per year in a solar greenhouse. Thus, seeking an appropriate method to delay leaf senescence and further unraveling the mechanisms regulating leaf senescence of cucumber plants will help extend their life and improve the harvest yield during cucumber production.

Leaf senescence is generally the first stage of plant senescence, during which a number of degenerative events occur, such as a decrease in metabolic activities, cell death, protein degradation, and decomposition of organoids [4,5,6]. Leaf senescence, characterized by yellowing leaves, results in a decline in photosynthetic efficiency and limits the growth and yield of plants, and it is often accompanied by age or accelerated by abiotic stress, such as chilling, high temperature, drought, and darkness [6,7,8]. Leaf senescence has been reported to be related to the accumulation of reactive oxygen species (ROS), which further results in aggravated oxidative damage to cellular macromolecules [9,10]. For instance, the overaccumulation of superoxide anion (O_2_^−^) and hydrogen peroxide (H_2_O_2_), caused by the upregulation of the expression of respiratory burst oxidase homolog (RBOH) genes, was observed in senescent leaves of Chinese flowering cabbage during storage after harvest [11]. In addition, phytohormone signals are also involved in the regulation of leaf senescence, such as abscisic acid (ABA), jasmonic acid (JA), salicylic acid (SA), auxin (IAA), and cytokinin (CTK), and among these phytohormones, the application of exogenous ABA, JA, and SA enhances leaf senescence, and higher endogenous contents of ABA, JA, and SA are observed in senescent leaves, while IAA and CTK are reported to positively delay leaf senescence [12,13,14,15,16].

Melatonin (MT) is a small indole substance that is ubiquitous in animals, prokaryotes, and plants. MT, as a new type of hormone-like substance, was first discovered in the bovine pineal gland in 1958 [17]. Numerous publications have reported that MT not only regulates root development and promotes the growth of plants [18] but is also a powerful antioxidant that can delay leaf senescence induced by age or abiotic stresses [19,20,21,22], and the main mechanism of melatonin is directly scavenging ROS or activating antioxidant enzymes by upregulating mRNA abundance and promoting the ascorbate–glutathione (AsA–GSH) cycle to eliminate excess ROS [23]. For instance, Shi reported that MT delayed dark-induced leaf senescence by improving antioxidant and AsA–GSH cycle systems to decrease the accumulation of ROS and inhibit the expression of *SAGs* [24]. Meanwhile, the downregulation of *RBOH* (*RbohB*, *RbohC*, *RbohD*, *RbohD2*, and *RbohE*) was observed in melatonin-treated Chinese flowering cabbage to inhibit RBOH-mediated ROS production [11]. In addition, regarding the specific mechanism of action, the function of MT in the regulation of leaf senescence is related to other phytohormones [25]. For instance, exogenous application of melatonin repressed the expression of ABA biosynthetic genes and ABA signal-related genes *ABF1*, *ABF4*, and *ABI5* to decrease ABA content and delayed cabbage senescence [26], and CTK was reported to be activated by melatonin to alleviate leaf senescence [7].

Thus, the application of MT may be an effective approach to delay leaf senescence and finally increase the yield in greenhouse vegetable production; however, the role of MT in regulating senescence is not as clear as that of other phytohormones. In our previous study, we found that MT significantly improved the chilling tolerance of cucumber by enhancing photosynthetic carbon assimilation [27]. In this study, we constructed an overexpression vector of the key MT biosynthetic gene *ASMT* and found that the overexpression of *ASMT* obviously upregulated the mRNA abundance of antioxidant enzyme genes and promoted the accumulation of AsA and GSH, which further decreased the excess ROS content in dark-induced senescent cucumber leaves. More importantly, we also found that the application of melatonin could also improve the chlorophyll content and downregulate *SAG20* and the cell death of leaves at 28, 35, and 42 days of leaf age, which ultimately delayed the leaf senescence of cucumber plants in the solar greenhouse. These results will be valuable for increasing yields in cucumber plants and will provide a theoretical basis for the high-efficiency application of MT during vegetable cultivation.

## 2. Results

### 2.1. Dynamic Changes in Endogenous MT in Cucumber Leaves in Which Senescence Was Induced by Darkness

To investigate the effect of senescence factors on the content of melatonin in cucumber leaves, a dark environment was used to induce rapid senescence. In cucumber leaves, the content of melatonin first increased and then decreased with the extension of treatment time. After 3 days of treatment, the content of melatonin increased to the highest point, 0.4723 ng·g^−1^, and then decreased to the lowest point, 0.1333 ng·g^−1^ (Figure 1A). In addition, we also found a change in the trend of mRNA abundance of *TDC*, *T5H*, *SNAT*, and *ASMT*, which are the key melatonin synthesis genes, and Figure 1B–E shows that these changes were consistent with the melatonin content. This suggests that darkness induces melatonin synthesis and indirectly affects melatonin content.

### 2.2. Effects of ASMT Overexpression on Dark-Induced Cucumber Leaf Senescence

To further understand the important role of MT in leaf senescence, we constructed a transient overexpression vector of *CsASMT*, a key gene for MT synthesis, and the endogenous MT content was indeed increased (data not shown). As shown in Figure 2A, after 5 days of dark stress, the cotyledons of cucumber seedlings in the control group showed yellow coloring and serious wilting. Moreover, the soluble protein content of cucumber seedlings in the control was significantly lower than that in the *OE-ASMT* seedlings, and there was a significant difference after treatment for 3 days. After treatment for 5 days, the soluble protein content of the control seedlings decreased by 180% compared with that in the *OE-ASMT* seedlings (Figure 2C). Overexpression of *CsASMT* also had a more obvious effect on alleviating MDA accumulation, the difference was obvious at the early stage of dark stress, and the difference increased with the extension of treatment time (Figure 2B). These results further demonstrated the positive regulation of MT on cucumber leaf senescence.

### 2.3. Effects of ASMT Overexpression on ROS Content in Cucumber Leaves in Which Senescence Was Induced by Darkness

As shown in Figure 3A, the H_2_O_2_ and O_2_^−^ contents of cucumber cotyledons in control and *CsASMT* overexpression plants increased gradually (Figure 3A,B), and after 5 days of darkness, the contents of H_2_O_2_ and O_2_^−^ in leaves of overexpressed *CsASMT* plants were significantly lower than those of the control, decreasing by 29.5% and 15.7%, respectively. In accordance with the change in ROS content, the diaminobenzidine (DAB) and nitroblue tetrazolium (NBT) staining (Figure 3C) results confirmed that ROS increased significantly in both treatments and that ROS content was higher than that in *CsASMT*-overexpressing plants under darkness.

### 2.4. Effects of ASMT Overexpression on Antioxidant Enzyme Activity in Cucumber Leaves in Which Senescence Was Induced by Darkness

To further explore the effect of *CsASMT* overexpression on the antioxidant defense system of cucumber under darkness, we further measured the activities and relative mRNA expression levels of superoxide dismutase (SOD) and catalase (CAT). The data showed that darkness induced the activities of SOD and CAT, and their activities in *CsASMT* overexpression seedlings were obviously higher than those in the control. Meanwhile, the overexpression of ASMT also increased the mRNA abundance of SOD and CAT compared with the control seedlings under dark conditions (Figure 4). The results implied that MT could effectively improve the activity and expression of antioxidant enzymes and further alleviate the accumulation of ROS.

### 2.5. MT Delays the Cucumber Leaf Senescence Induced by Age

To further verify whether the application of MT could be used during cucumber production, we observed the leaf phenotype and measured the change in MDA and soluble protein contents in leaves of different ages. As shown in Figure 5, the leaves of cucumber were obviously yellow-green, showed obvious signs of senescence at 42 days of age, and showed a significant increase in MDA content and a decrease in soluble protein content. However, the application of MT notably showed more green leaves, lower MDA content, and higher protein contents compared with the H_2_O treatment. For instance, the soluble protein content was 31.4% higher than that of the control at 42 days of leaf age.

### 2.6. MT Regulates Chlorophyll Metabolism Enzyme Activity to Maintain Chlorophyll Content during Cucumber Leaf Senescence

Yellowing is a notable phenomenon during leaf senescence. To prove whether MT could relieve the decrease in chlorophyll during leaf senescence induced by age, we observed changes in not only the activity of chlorophyll synthetase but also chlorophyll degradation enzymes. Figure 6A shows that the chlorophyll contents of cucumber leaves first increased and then decreased at the leaf age of 35 days, which implied that the leaves began to be senescent. Meanwhile, we found that the activities of PAO, PPH, MgCH, and FeCH showed a similar trend in the chlorophyll contents with increasing leaf age; however, the application of MT maintained higher MgCH and FeCH activities, which were related to chlorophyll synthesis, and suppressed PAO and PPH activities, which were related to chlorophyll degradation (Figure 6C–F). These results signify the importance of MT in regulating the balance of chlorophyll synthesis and degradation to delay leaf senescence in cucumber.

### 2.7. MT Downregulates Cell-Death- and Senescence-Related Gene mRNA Abundance to Delay Leaf Senescence Induced by Age

During the leaf senescence process, *SAG* mRNA levels were significantly upregulated, and plants started cell death. Here, we found that leaf age induced the relative mRNA expression of *SAG20* and *PDCD* (Figure 7A,B), which gradually increased with increasing leaf age. Importantly, MT apparently alleviated cell death and downregulated the *SAG20* mRNA level compared with the H_2_O-treated plants. The staining of programmed cell death was in agreement with the change in *PDCD* mRNA levels (Figure 7C).

### 2.8. MT Modulates IAA and ABA Signals and Synthesis to Delay Leaf Senescence Induced by Age

Considering the importance of phytohormones for leaf senescence, we measured the mRNA abundance of IAA- and ABA-related genes in both H_2_O- and MT-treated leaves at different leaf ages. With the increase in age, *ARF1* and *YUCCA6* mRNA abundance levels first increased and then decreased at the leaf age of 35 days, when leaves began to be senescent; however, *ARF1* and *YUCCA6* mRNA abundance levels in MT were notably higher than those of H_2_O treatment (Figure 8A,B), implying that MT could significantly alleviate the decrease in IAA in age-induced senescent leaves. In contrast to the change in IAA, the mRNA abundance of the ABA-related genes *ABI5* and *NCED* increased during the leaf development process (Figure 8C,D), but there was no significant difference between the H_2_O and MT treatments before the leaf age of 28 days, and *ABI5* and *NCED* mRNA abundance levels in the MT treatment were obviously lower than those in the H_2_O treatment after the leaf age of 35 days, implying that MT inhibited ABA synthesis and signal transduction in senescent cucumber leaves.

### 2.9. MT-Delayed Leaf Senescence Is Associated with Increased ROS-Scavenging Activity 

Through the measurement of the H_2_O_2_ content and O_2_^−^ production rate in leaves during natural senescence within 42 days or in leaves in which senescence was induced by darkness, it can be seen that the contents of H_2_O_2_ and the O_2_^−^ production rate in leaves gradually increased during senescence induced by age or darkness, but the application of exogenous MT significantly inhibited the accumulation of ROS in both experiments compared with the H_2_O treatment (Figure 9).

Plants activate the antioxidant system when excess ROS occur during senescence or under abiotic stresses. Here, we observed the effect of exogenous MT on the activities and the relative expression of antioxidant enzymes during the cucumber leaf senescence process. As shown in Figure 10, the application of MT significantly promoted the activities of SOD, CAT, and ascorbate peroxidase (APX) in leaves in which senescence was induced by age or darkness. Moreover, real-time PCR data showed that senescence induced the mRNA expression of *SOD*, *CAT*, and *APX,* and MT-treated seedlings displayed obviously higher relative mRNA expression of *SOD*, *CAT*, and *APX* than H_2_O-treated seedlings (Figure 11), indicating that MT could upregulate the mRNA abundance of genes to further activate antioxidant enzyme activity. Considering that redox species are involved in ROS scavenging, we also examined the change in ASA and GSH contents of leaves at different ages or treated with darkness sprayed with or without MT. Our results showed that leaf senescence notably increased the contents of ASA and GSH, and this effect of MT on redox species contents was largely higher than that of H_2_O treatment in leaves in which senescence was induced by age or darkness (Figure 12), which implied that MT could promote the antioxidant capacity to scavenge ROS accumulation caused by senescence induced by age or darkness.

## 3. Discussion

MT serves as an important signaling molecule in the regulation of leaf senescence induced by abiotic stresses, darkness, and age [19,23,28]. With increasing age, the endogenous MT content of Arabidopsis thaliana increased, and the application of exogenous MT also increased the concentration of MT to further delay leaf senescence [19], implying that the endogenous MT level played a vital role in the regulation of MT on the resistance of plants. *TDC*, *T5H*, *SNAT*, and *ASMT* or *HIOMT* were the key genes for MT synthesis, which was edited as a genetic approach to change the MT level [29]. For instance, the overexpression of *ASMT* significantly alleviated photoinhibition and induced the expression of HSPs to promote the thermotolerance of tomato [30], while the silencing of *TDC* aggravated the accumulation of cadmium in tomato [31]. Meanwhile, *COMT1* overexpression promoted melatonin biosynthesis and contributed to the alleviation of carbendazim phytotoxicity and residues in tomato plants [32]. Usually, leaf senescence induced by darkness is used to simulate natural senescence because leaves with dark-induced senescence show changes in chlorophyll content and a yellow leaf phenotype similar to those during natural senescence [33,34]. Thus, we first studied the effect of ASMT overexpression on leaf senescence induced by darkness, and the data showed that darkness induced the upregulation of *CsT5H*, *CsASMT*, and *CsTDC* mRNA abundance, which further led to an increase in endogenous MT content, implying that MT acts as a regulator during leaf senescence (Figure 1). Overexpression of *ASMT* dramatically delayed dark-induced leaf senescence compared with the control, as evidenced by the green phenotype and lower contents of MDA and soluble protein (Figure 2). Consistent with this result, according to the data from the field experiment, the application of exogenous MT also notably decreased the MDA and soluble protein contents in senescing cucumber leaves at 28, 35, and 42 days of leaf age in the solar greenhouse (Figure 5).

Previous studies have demonstrated that ROS are involved in the regulation of plant senescence [35,36,37], and excess ROS content leads to oxidative damage to lipids, proteins, and DNA during leaf senescence [38,39]. In this paper, the quantification and histochemical analysis of H_2_O_2_ and O_2_^−^ showed that obvious ROS build-up was observed in senescing cucumber leaves induced by darkness or age, which was lower in *ASMT*-overexpressing and melatonin-treated cucumber leaves (Figure 3 and Figure 9), and thus, the lower lipid peroxidation product MDA content was also measured in *ASMT*-overexpressing or MT-treated cucumber leaves, which was in accordance with the results of Tan [11] in Chinese cabbage, and melatonin was thought to be an antioxidant [39,40]. Usually, leaf senescence is considered to be a form of programmed cell death (PCD), and higher transcript levels of senescence-associated genes (*SAGs*) are induced during this process [4,5]. Programmed cell death is associated with abiotic-triggered oxidative stress caused by excessive production of ROS, such as H_2_O_2_ and O_2_^−^, in cells [41,42]. Our study showed that a corresponding increase in cell death in both trypan blue staining and *PDCD* mRNA abundance and an upregulation of *SAG12* mRNA abundance were observed in senescing cucumber leaves. However, aging cucumber leaves pretreated with melatonin effectively inhibited cell death, which was related to the lower ROS content observed in senescing cucumber leaves induced by darkness or age (Figure 3, Figure 7 and Figure 9). The yellowing of green leaves due to the degradation of chlorophyll (Chl) and Chl–protein complexes is the first observed typical phenomenon during leaf senescence [43], and chlorophyllase1 (CLH1), CLH2, pheophytin pheophorbide hydrolase (PPH), pheophorbide A oxygenase (PAO), and NYC1-like (NOL) play a vital role in chlorophyll degradation, and these are obviously upregulated by senescence-related phytohormones as well as abiotic stresses to accelerate leaf senescence [44,45,46,47,48]. Here, we found a green phenotype and higher chlorophyll content by not only downregulating the activities of PAO and PPH but also activating the activities of MgCH and FeCH, which are the key enzymes contributing to the synthesis of chlorophyll in 28-, 35-, and 42-day-old cucumber leaves in a solar greenhouse (Figure 6). In addition, previous studies have revealed that the activation of the antioxidant defense system is the main mechanism by which MT regulates abiotic stress and leaf senescence [49,50,51]. Indeed, higher mRNA abundance and activities of SOD and CAT were measured in *ASMT*-overexpressing plants than in control plants after darkness (Figure 4). Similarly, the application of MT also induced the upregulation of SOD, CAT, and APX transcripts and activities as well as the AsA–GSH cycle (Figure 10, Figure 11 and Figure 12) in cucumber leaves compared with H_2_O-treated cucumber leaves during senescence induced by age or darkness, which was in accordance with the report of Tan [11].

IAA was reported to delay leaf senescence [15], and the differentially expressed genes from transcriptomic data of leaf senescence induced by darkness also proved evidence that IAA was involved in the regulation of senescence [52]. Elevated auxin levels caused by overexpression of key auxin synthesis genes, such as YUCC6, or application of exogenous IAA obviously delayed leaf senescence [53,54]. Furthermore, the change in the transcriptional level of IAA signaling transcription factors, such as AUXIN RESPONSE FACTOR 2 (ARF2) and AUXIN RESISTANT 3 (AXR3)/INDOLE-3-ACETIC ACID INDUCIBLE 17 (IAA17), also affected leaf senescence [19,55]. Contrary to the role of IAA, a high concentration of ABA accelerates leaf senescence [56]. Different lines of evidence highlight that ABA content or signaling is promoted or activated by leaf senescence induced by age or abiotic stress [7,57]. Consistent with the results of previous studies, our data also showed that the application of MT significantly activated the biosynthesis and signaling pathways of IAA while inhibiting the biosynthesis and signaling genes involved in ABA (Figure 8) in cucumber leaves in which senescence was induced by age, which further decreased the generation of ROS [39,58].

In summary, MT induced chilling tolerance in cucumber seedlings, as shown by the decrease in stress-induced electrolyte leakage, the decreased contents of H_2_O_2_ and MDA, and the production rate of O_2_^−^, which occurred partially due to the induction of antioxidant metabolism. On the other hand, MT treatment maintained a high photosynthetic carbon assimilation capacity and increased the activity of the PSII reaction center and electron transfer efficiency, thus alleviating the damage to the photosynthetic apparatus under chilling stress and increasing the chilling tolerance of cucumber seedlings.

In summary, the endogenous MT response to leaf senescence is induced by darkness. The overexpression of ASMT or the application of MT significantly increased antioxidant capacity to decrease the contents of MDA and H_2_O_2_ and the production rate of O_2_^−^, adjusted the balance of chlorophyll synthesis and degradation to increase the chlorophyll content, alleviated the decrease in IAA signaling and synthesis, decreased ABA signaling and synthesis, and finally delayed the leaf senescence induced by darkness or aging, as evidenced by lower cell death and *SAG20* mRNA abundance.

## 4. Materials and Methods

### 4.1. Cucumber Transient Transformation

The overexpression vector transformation and cucumber transient transformation were performed according to the method of Meng [59]. Briefly, the CDS region of *CsASMT* was subcloned into the pBI121 vector, and then the recombinant plasmid was transformed into *A. tumefaciens* strain EHA105. Moreover, EHA105 cells containing the pBI121-*CsASMT* plasmid were cultured to OD600 = 0.8, which was diluted to OD600 = 0.4 when the bacterial solution was injected into 7-day-old cucumber cotyledons using a syringe without a needle. After 24 h of darkness, the cucumber seedlings were transferred to normal conditions (day/night 25 °C/18 °C, 600 μmol m^−2^·s^−1^ PFD) for 1 day. Then, the seedlings were treated with darkness and sampled at 0, 1, 3, and 5 days.

For the field experiment, cucumber plants (*Cucumis sativus* L. ‘Jinyou 35’) were planted in a solar greenhouse at the two-leaf stage and were routinely managed. The leaves were labeled at 0, 7, 14, 21, 28, 35, and 42 days of leaf age, and the day on which the four leaves emerged from the growing tip was recorded, and 100 μmol·L^−1^ MT was sprayed every 7 days at the four-leaf stage. The plants treated with H_2_O were used as the control. Then, we sampled 0-, 7-, 14-, 21-, 28-, 35-, and 42-day-old leaves for the following analysis. To avoid leaf senescence caused by mutual shade between leaves, cucumber plants were grown close to the ground without hanging.

### 4.2. Measurement of MDA, Soluble Protein, and Chlorophyll Contents

MDA content was estimated with the thiobarbituric acid (TBA) colorimetric method as described by [60]. The soluble protein and chlorophyll contents were measured according to the method reported by Zhao [61].

### 4.3. Key Chlorophyll Degradation Enzyme Activities

The activities of PPH and PAO enzymes were determined according to the instructions of the kits (H-60393, H-60401) manufactured by Suzhou Keming Biotechnology Co., Ltd., Suzhou, China; MgCH and FeCH enzyme activities were determined according to the instructions of the kit produced by Suzhou Keming Biotechnology Co., Ltd., Suzhou, China.

### 4.4. Measurement of ROS Content

H_2_O_2_ and superoxide anion (O_2_^−^) contents were measured at 0, 1, 3, and 5 days after darkness and at 7, 14, 21, 28, 35, and 42 days of leaf age in cucumber leaves in a solar greenhouse. The quantitation of H_2_O_2_ content was determined by the plant H_2_O_2_ Assay Kit (A064-1, Nanjing Jiancheng Bioengineering Institute of China, Nanjing, China). The O_2_^−^ production rate was determined using the method described by Wang [62].

Cellular H_2_O_2_ was detected with an inverted fluorescence microscope using the H_2_O_2_ fluorescent probe 2′,7′-dichlorodihydrofluorescein diacetate (H_2_DCFDA) (MCE, Cat. No. HY-D0940, Shanghai, China), and O_2_^−^ was visualized with dihydroethidium (DHE) (Fluorescence Biotechnology Co. Ltd., Cat. No. 15200, Beijing, China) according to our previous paper [63,64].

### 4.5. NBT and DAB Staining

NBT staining of O_2_^−^ was performed according to the method of Jabs et al. (1996) [65] with minor modifications. The fresh leaves were washed with distilled water, immersed in 0.5 mM NBT in a vacuum, and stained at 28 °C for 1 h. Then, the leaves were boiled in a mixed ethanol:lactic acid:glycerol (3:1:1) solution to remove pigments, and O_2_^−^ was visualized by blue-purple coloration. DAB staining of H_2_O_2_ was carried out as described by Thordal-Christensen et al. (1997) [66]. The cleaned fresh leaves were soaked in 1 mM DAB staining solution (pH 3.8) in a vacuum and stained at 28 °C for 8 h. Then, the leaves were boiled in a mixed ethanol:lactic acid:glycerol (3:1:1) solution to remove pigments, and H_2_O_2_ was visualized by reddish-brown coloration.

### 4.6. Antioxidant Enzyme Activity Assay

Superoxide dismutase (SOD) activity was measured according to Beyer and Fridovich’s method [67]. The CAT activity was measured according to the method described by Chance [68]. The ascorbate peroxidase (APX) activity was measured according to the method described by Nakano and Asada [69], and the catalase (CAT) activity was assayed in accordance with the method of Li [70].

### 4.7. Determination of the Redox Substance Contents

The GSH content was measured according to the instructions of the GSH-2-W kit (Suzhou KeMing Bioengineering Institute, Suzhou, China). The AsA content was tested according to the method described by Li et al. [71].

### 4.8. Trypan Blue Staining for Cell Death

Different leaf-age cucumber leaves were collected for the observation of cell death and made into discs 1.5 cm in diameter. Then, the discs were transferred to trypan blue staining solution (2.5 mg/mL trypan blue, 25% lactic acid, 23% water phenol, 25% glycerinum) according to the method of Yin et al. [72].

### 4.9. Measurements of MT Content

Cucumber leaves treated with H_2_O and aged 7, 14, 21, 28, 35, and 42 days were taken and immediately frozen in liquid nitrogen. The frozen leaves were dried in a vacuum freeze dryer and ground into frozen powder under liquid nitrogen freezing, and 0.1 g of powder was weighed for MT extraction [27]. A 10 μL extract was then tested on an HPLC–MS system (Thermo Fisher Scientific, TSQ Quantum Access, Waltham, MA, USA) with the test conditions set according to Bian [73].

### 4.10. RNA Extraction and Gene Expression Analysis

An RNA TRIzol kit was used for total RNA extraction from sample cucumber leaves (Tiangen, Beijing, China), and cDNA was obtained according to the instructions of the reverse transcription system. For real-time quantitative PCR (RT-qPCR), the reaction was performed with a TransStart TipTop Green qPCR Super Mix (Cwbio, Beijing, China) using a LightCycler 480 II system (Roche, Penzberg, Germany), and the cucumber ß-actin gene (XM_011659465) was used as an internal reference gene. The RT-qPCR primers are shown in Table 1.

### 4.11. Statistical Analysis

The experiments were designed with at least three replicates, which contained 5–10 plants. The data are presented as the mean ± the standard deviation (SD). All data were analyzed statistically using DPS software. Statistical analysis of the values was performed by Duncan’s multiple range test (DMRT), and comparisons with *p* < 0.05 were considered significantly different.

## Figures and Tables

**Figure 1 ijms-23-03576-f001:**
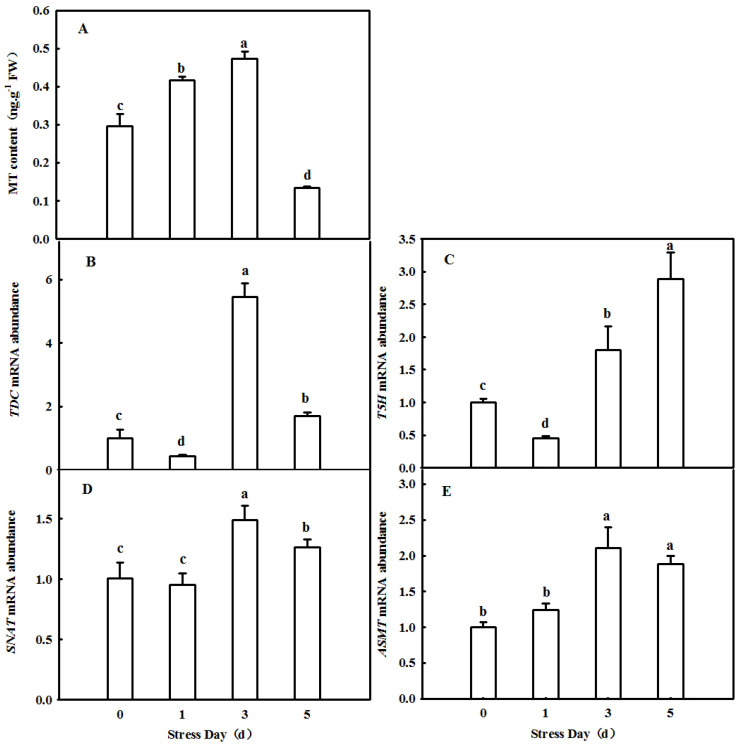
The variation of endogenous MT (**A**) and mRNA expression of *TDC* (**B**), *T5H* (**C**), *SNAT* (**D**), and *ASMT* (**E**) in leaves of cucumber plants under darkness. The two-leaf stage cucumber seedlings were treated with dark condition (day/night temperature: 25 °C/18 °C). The 2nd leaf samples were taken at 0, 1, 3, and 5 days. All values shown are mean ± SD (*n* = 3). a–d indicate that mean values are significantly different among samples (*p* < 0.05).

**Figure 2 ijms-23-03576-f002:**
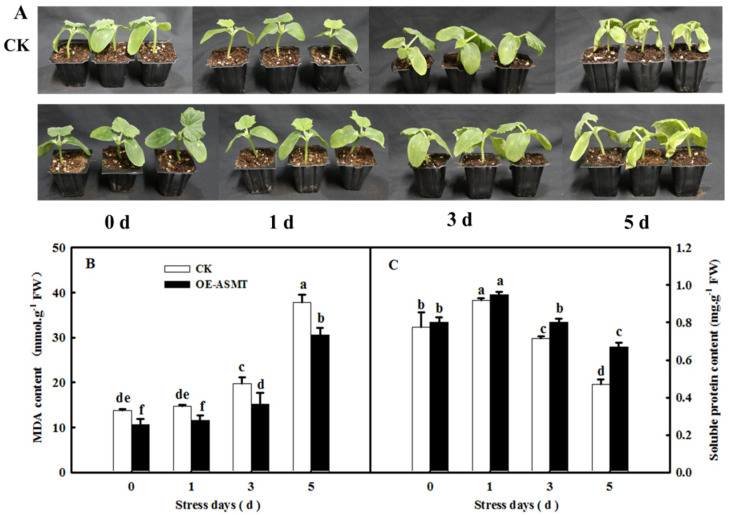
The effect of *CsASMT* transient overexpression on senescence of leaves under dark condition. (**A**) The phenotype of cotyledons; (**B**) MDA content; (**C**) soluble protein content. The *CsASMT* transient overexpression plants with two cotyledons were displaced into growth chambers and treated with dark condition (day/night temperature: 25 °C/18 °C). The cotyledons were collected for the determination of MDA and soluble protein contents at 0, 1, 3, and 5 days. All values shown are mean ± SD (*n* = 3). a–f indicate that mean values are significantly different among samples (*p* < 0.05).

**Figure 3 ijms-23-03576-f003:**
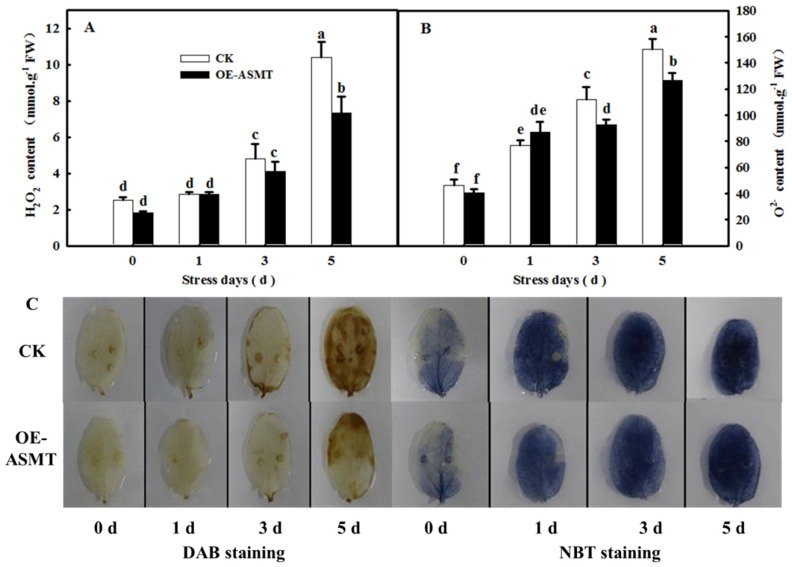
The effect of *CsASMT* transient overexpression on ROS content of leaves under dark condition. (**A**) H_2_O_2_ content; (**B**) O_2_^−^ content; (**C**) DAB and NBT staining. The *CsASMT* transient overexpression plants with two cotyledons were displaced into growth chambers and treated with dark condition (day/night temperature: 25 °C/18 °C). The cotyledons were collected for ROS content determination at 0, 1, 3, and 5 days. All values shown are mean ± SD (*n* = 3). a–f indicate that mean values are significantly different among samples (*p* < 0.05).

**Figure 4 ijms-23-03576-f004:**
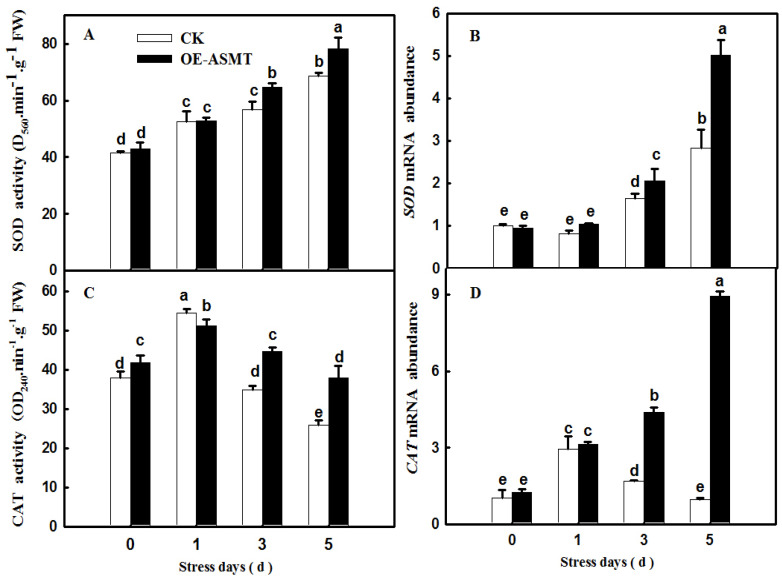
The effect of *CsASMT* transient overexpression on the activities and relative mRNA expressions of antioxidant enzyme in cucumber leaves under dark condition. (**A**) SOD activity; (**B**) the relative expression of *SOD*; (**C**) CAT activity; (**D**) the relative expression of *CAT*. The *CsASMT* transient overexpression plants with two cotyledons were displaced into growth chambers and treated with dark condition (day/night temperature: 25 °C/18 °C). The cotyledons were collected for the determination of antioxidant enzyme activity and related mRNA at 0, 1, 3, and 5 days. All values shown are mean ± SD (*n* = 3). a–e indicate that mean values are significantly different among samples (*p* < 0.05).

**Figure 5 ijms-23-03576-f005:**
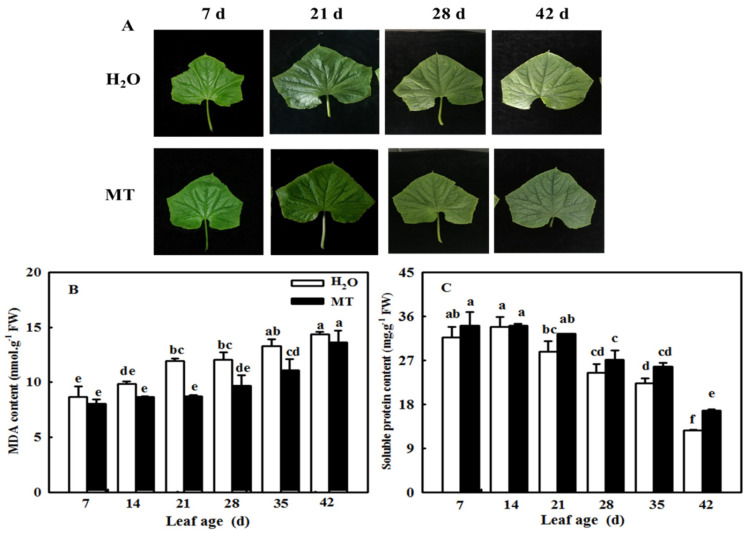
The effect of exogenous MT on MDA and protein contents in different age leaves of cucumber plants in a solar greenhouse. (**A**) Phenotype of leaves at different ages; (**B**) MDA content; (**C**) soluble protein content. At the three-leaf stage, the new leaf at the top was labeled 0 days and then labeled at 7, 14, 21, 28, 35, and 42 days. Meanwhile, cucumber plants were sprayed with 100 μmol·L^−1^ MT every 7 days, and plants treated with H_2_O were the control. The different leaf-age leaves were sampled for MDA and protein content determination. All values shown are mean ± SD (*n* = 3). a–f indicate that mean values are significantly different among samples (*p* < 0.05).

**Figure 6 ijms-23-03576-f006:**
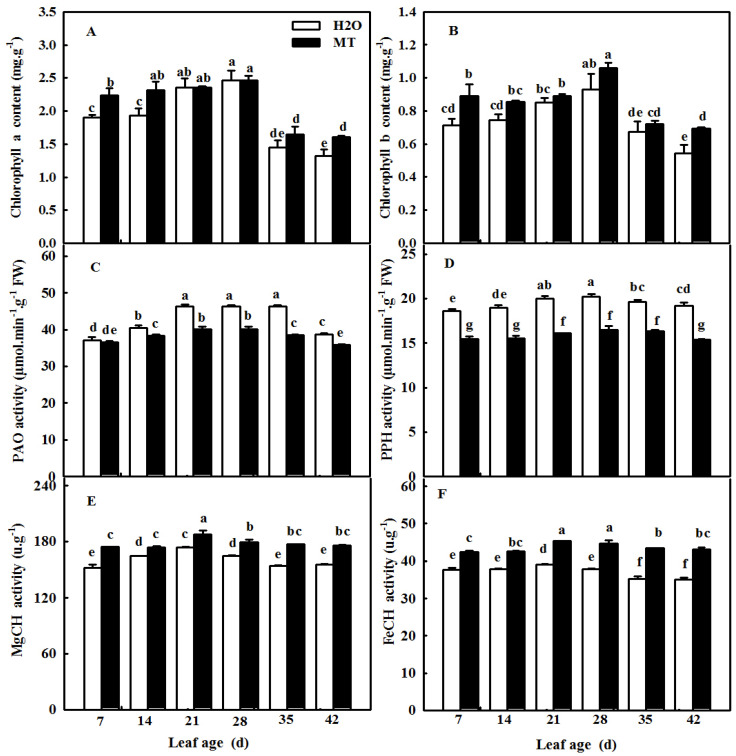
The effect of exogenous MT on chlorophyll metabolism in different age leaves of cucumber plants in a solar greenhouse. (**A**) Chlorophyll a content; (**B**) chlorophyll b content; (**C**) PAO activity; (**D**) PPH activity; (**E**) MgCH activity; (**F**) FeCH activity. At the three-leaf stage, the new leaf at the top was labeled 0 days and then labeled at 7, 14, 21, 28, 35, and 42 days. Meanwhile, cucumber plants were sprayed with 100 μmol·L^−1^ MT every 7 days, and plants treated with H_2_O were the control. The different leaf-age leaves were sampled for the determination of chlorophyll contents and relative enzyme activities. All values shown are mean ± SD (*n* = 3). a–g indicate that mean values are significantly different among samples (*p* < 0.05).

**Figure 7 ijms-23-03576-f007:**
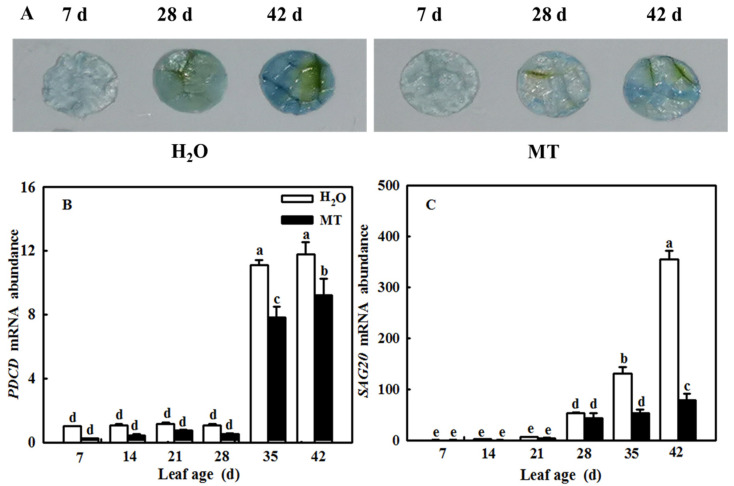
The effect of exogenous MT on cell-death- and senescence-related gene mRNA abundance in different age leaves of cucumber plants in a solar greenhouse. (**A**) The stain of programmed cell death; (**B**) *PDCD* mRNA abundance; (**C**) *SAG20* mRNA abundance. At the three-leaf stage, the new leaf at the top was labeled 0 days and then labeled at 7, 14, 21, 28, 35, and 42 days. Meanwhile, cucumber plants were sprayed with 100 μmol·L^−1^ MT every 7 days, and plants treated with H_2_O were the control. The different leaf-age leaves were sampled for trypan blue staining and gene mRNA abundance determination. All values shown are mean ± SD (*n* = 3). a–e indicate that mean values are significantly different among samples (*p* < 0.05).

**Figure 8 ijms-23-03576-f008:**
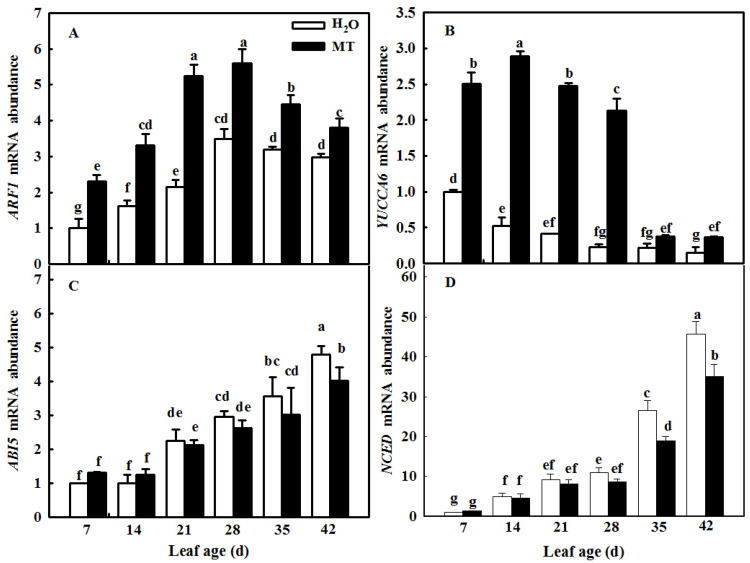
The effect of exogenous MT on IAA-/ABA-related gene mRNA abundance in different age leaves of cucumber plants in a solar greenhouse. (**A**) *ARF1* mRNA abundance; (**B**) *YUCCA6* mRNA abundance; (**C**) *ABI5* mRNA abundance; (**D**) *NCED* mRNA abundance. At the three-leaf stage, the new leaf at the top was labeled 0 days and then labeled at 7, 14, 21, 28, 35, and 42 days. Meanwhile, cucumber plants were sprayed with 100 μmol L^−1^ MT every 7 days, and plants treated with H_2_O were the control. The different leaf-age leaves were sampled for gene mRNA abundance determination. All values shown are mean ± SD (*n* = 3). a–g indicate that mean values are significantly different among samples (*p* < 0.05).

**Figure 9 ijms-23-03576-f009:**
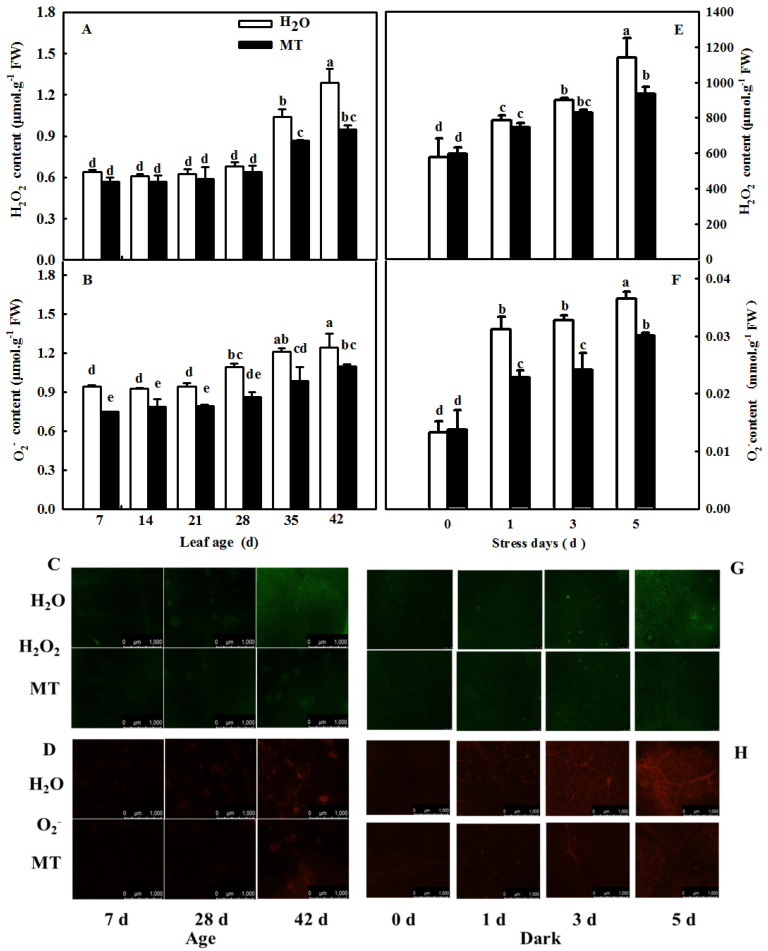
The effect of exogenous MT on ROS content in senescent leaves of cucumber plants induced by age or darkness. (**A**) H_2_O_2_ content in different age leaves; (**B**), O_2_^−^ content in different age leaves; (**C**) inverted microscope imaging of H_2_O_2_ in different age leaves; (**D**) inverted microscope imaging of O_2_^−^ in different age leaves (**E**) H_2_O_2_ content in leaves under darkness; (**F**) O_2_^−^ content in leaves under darkness; (**G**) inverted microscope imaging of H_2_O_2_ in leaves under darkness; (**H**) inverted microscope imaging of O_2_^−^ in leaves under darkness. At the two-leaf stage, the cucumber seedlings were planted in a solar greenhouse or started to be treated with MT. At the three-leaf stage, the new leaf at the top of cucumber plants was labeled 0 days and then labeled at 7, 14, 21, 28, 35, and 42 days in a solar greenhouse; plants were sprayed with 100 μmol·L^−1^ MT every 7 days, and plants treated with H_2_O were the control. Meanwhile, the seedlings were treated with H_2_O and 100 μmol·L^−1^ MT, respectively, 2 times, and then seedlings were displaced into growth chambers and treated with dark condition (day/night temperature: 25 °C/18 °C). The 2nd leaf samples were taken at 0, 1, 3, and 5 days. All values shown are mean ± SD (*n* = 3). a–e indicate that mean values are significantly different among samples (*p* < 0.05).

**Figure 10 ijms-23-03576-f010:**
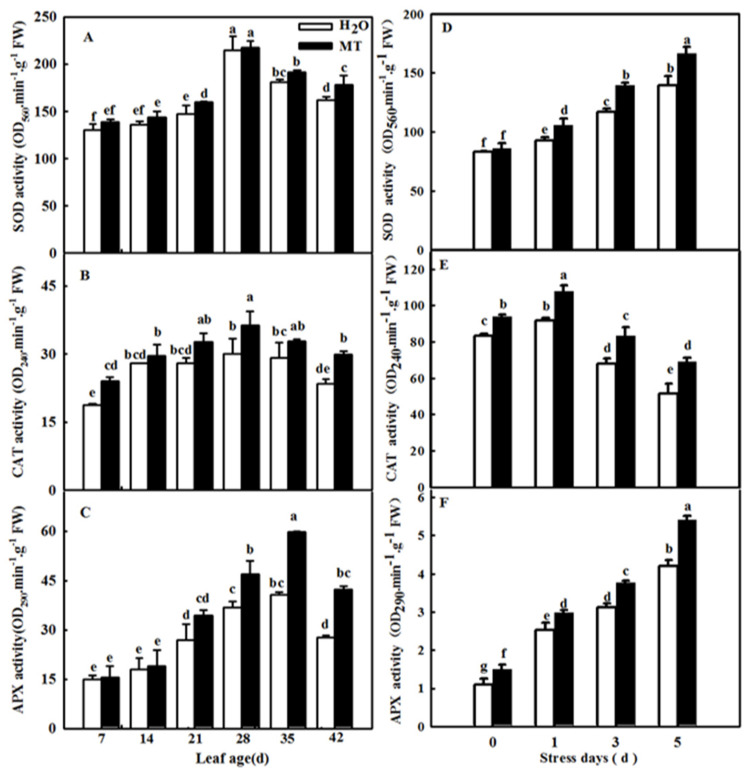
The effect of exogenous MT on antioxidant enzyme activity in senescent leaves of cucumber plants induced by age or darkness. (**A**–**C**) SOD, CAT, and APX activities in different age leaves; (**D**–**F**) SOD, CAT, and APX activities in leaves under darkness. At the two-leaf stage, the cucumber seedlings were planted in a solar greenhouse or started to be treated with MT. At the three-leaf stage, the new leaf at the top of cucumber plants was labeled 0 days and then labeled at 7, 14, 21, 28, 35, and 42 days in a solar greenhouse; plants were sprayed with 100 μmol·L^−1^ MT every 7 days, and plants treated with H_2_O were the control. Meanwhile, the seedlings were treated with H_2_O and 100 μmol·L^−1^ MT, respectively, 2 times, and then seedlings were displaced into growth chambers and treated with dark condition (day/night temperature: 25 °C/18 °C). The 2nd leaf samples were taken at 0, 1, 3, and 5 days. All values shown are mean ± SD (*n* = 3). a–g indicate that mean values are significantly different among samples (*p* < 0.05).

**Figure 11 ijms-23-03576-f011:**
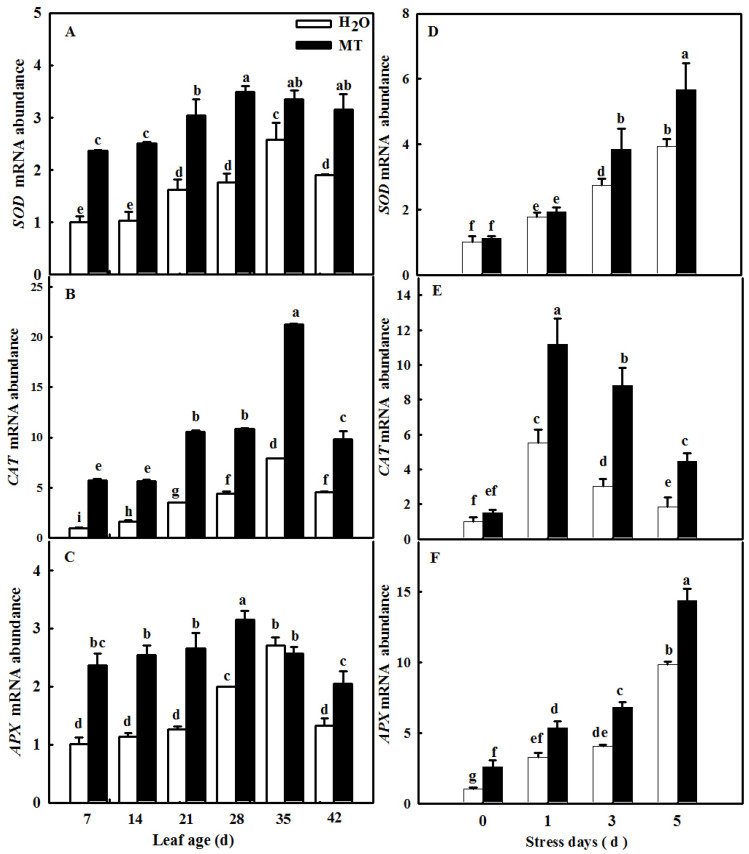
The effect of exogenous MT on the mRNA abundance of antioxidant enzyme genes in senescent leaves of cucumber plants induced by age or darkness. (**A**–**C**) *SOD*, *CAT*, and *APX* mRNA abundance in different age leaves; (**D**–**F**) *SOD*, *CAT*, and *APX* mRNA abundance in leaves under darkness. At the two-leaf stage, the cucumber seedlings were planted in a solar greenhouse or started to be treated with MT. At the three-leaf stage, the new leaf at the top of cucumber plants was labeled 0 days and then labeled at 7, 14, 21, 28, 35, and 42 days in a solar greenhouse; plants were sprayed with 100 μmol·L^−1^ MT every 7 days, and plants treated with H_2_O were the control. Meanwhile, the seedlings were treated with H_2_O and 100 μmol·L^−1^ MT, respectively, 2 times, and then seedlings were displaced into growth chambers and treated with dark condition (day/night temperature: 25 °C/18 °C). The 2nd leaf samples were taken at 0, 1, 3, and 5 days. All values shown are mean ± SD (*n* = 3). a–i indicate that mean values are significantly different among samples (*p* < 0.05).

**Figure 12 ijms-23-03576-f012:**
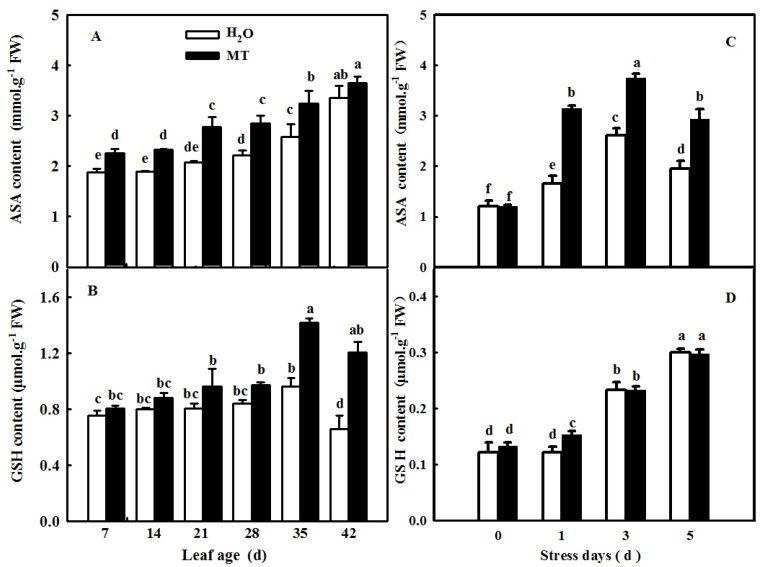
The effect of exogenous MT on the ASA and GSH contents in senescent leaves of cucumber plants induced by age or darkness. (**A**,**B**) ASA and GSH contents in different age leaves; (**C**,**D**) ASA and GSH contents in leaves under darkness. At the two-leaf stage, the cucumber seedlings were planted in a solar greenhouse or started to be treated with MT. At the three-leaf stage, the new leaf at the top of cucumber plants was labeled 0 days and then labeled at 7, 14, 21, 28, 35, and 42 days in a solar greenhouse; plants were sprayed with 100 μmol·L^−1^ MT every 7 days, and plants treated with H_2_O were the control. Meanwhile, the seedlings were treated with H_2_O and 100 μmol·L^−1^ MT, respectively, 2 times, and then seedlings were displaced into growth chambers and treated with dark condition (day/night temperature: 25 °C/18 °C). The 2nd leaf samples were taken at 0, 1, 3, and 5 days. All values shown are mean ± SD (*n* = 3). a–f indicate that mean values are significantly different among samples (*p* < 0.05).

**Table 1 ijms-23-03576-t001:** The primers for RT-PCR.

Genes	Accession Numbers	Primer Pairs (5′–3′)
*ASMT*	XM_004144879	ATTGGAAGTTTAGTTGATGTGGGA
		AGCATCAGCCTTGGGAATGGAAT
*TDC*	XM_004135488	ATAAATGGTTCTTCTCGGCGCCAG
		GTTAATCATATTCGACTTCTGGT
*T5H*	XM_004140201	AGCTTGTGCAGGCTACCAACT
		GAACGTTGGAACAAACTTGTG
*SNAT*	XM_011655429	AGTCCCCTGTTTCAGAGGAGAAT
		AGATTCCGATAAAACTCTACCAC
*SAG20*	XM_004149882	CAGACCTGGAGTGGTGGTTC
		GCCGGAGATCTGTCACAACA
*PDCD*	XM_011661791	AGATGATGATGACGACGATG-
		CAGCCTTGCTTGGAAATAG
*PAO*	XM_031884976	GGGCATTGAAAACTGGAAGA
		TTACTTGGCGATCAAAAATGG
*PPH*	XM_011661125	GCAATGTGACGCCCTTAACT
		CATCGAACAGGTCATTGGTG
*SOD*	XM_011660217	GGAAAGATGTGAAGGCTGTGG
		GCACCATGTTGTTTTCCAGCAG
*CAT*	XM_001308916	AATGGCCGGAGGATGTGA
		CCAACGACATAGAGAAAGCCAAG
*APX*	XM_001280706	GTGCTACCCTGTTGTGAGTG
		AACAGCGATGTCAAGGCCAT
*ARF1*	XM_011656213	CCAGATCCTCCCCTTCCTGA
		GTCATCCGCATGCCTCCTAA
*YUCCA6*	XM_004150231	GGGACACTGCAAGATTCGGA
		GCTTGACGTTTCAGCCGTTT
*NCED*	XM_004147720	TGGTGAACCGAAATCTACTTG
		CGAAGGCTAAGATGTGGC
*ABI5*	XM_011651278	GGAATTGCTTTTCAGCGGCA
		ACTCCATTGGCATTCAGCGA

## Data Availability

The original contributions presented in the study are included in the article, further inquiries can be directed to the corresponding authors.

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
