# Peer review of "Melatonin Positively Regulates Both Dark- and Age-Induced Leaf Senescence by Reducing ROS Accumulation and Modulating Abscisic Acid and Auxin Biosynthesis in Cucumber Plants"

_ijms, 2022, doi:10.3390/ijms23073576_

Round 1

Reviewer 1 Report

Overall, the work is interesting, and reports are still lacking about the influence of melatonin on plant ontogenesis.

However, it is worth remembering that in the case of plants, it is good not only to measure the expression level of a given gene but also the level of the corresponding protein. Both are often not directly correlated in plants, and inferring from this phenomenon may be wrong. The authors should refer to this issue in describing the results or discussions.

I have trouble understanding the results from the fluorescence microscope. At first, they look like background noise, turning up the power of the light source. Were the "leaves" used discs, or were leaves without epidermis? How was it separated from the autofluorescence of chlorophyll in the case of red fluorescence? Have any deconvolution algorithms been used? Can Authors be tempted to show a quantitative change in the intensity of pixels in a specific area? I am asking for explanations and possible improvements here.

After appropriate corrections, the work is ready for publication.

Reviewer 2 Report

Review of the manuscript entitled: Melatonin Positively Regulates Both Dark- and Age-Induced Leaf Senescence by Reducing ROS Accumulation and Modulating Abscisic Acid and Auxin Biosynthesis in Cucumber Plants

The manuscript describes the role of melatonin in dark- or age - induced leaf senescence and the link between the role of melatonin and ROS or plant hormones. While the role of melatonin in leaf senescence is well known and in this respect the work is not innovative, the precise recognition of these phenomena in a commercially significant species could be economically important. The paper describes two types of experiments conducted on plants treated with melatonin (sprayed) and transformed plants. However, throughout the paper I found no confirmation that the transformation was successful - and if indeed the authors were working with a plant overexpressing the gene CsASMT. I consider this to be a major shortcoming in the work and the lack of confirmation of a successful transformation prevents confidence in the results and puts into question the results obtained. In many places the style of language is strange, making it difficult to understand, in my opinion, the work needs language improvements. The authors do not state which statistical tests they use to assess the significance of the differences and whether the assumptions of these tests are met; they only state the software they use.

List of comments:

line 37 - this is a big shortcut, leaf senescence is not necessarily the last phase of plant growth - this is not the case with perennial plants

line 59: is: activating the activities of antioxidant enzymes, better will be: activating the antioxidant enzymes

The authors repeatedly use: meanwhile which sounds odd

All abbreviations, including those for genes and proteins, should be explained when first used and should be consistent, yet line 49 is CTK and line 70 is CT

Starting from line 126 the authors misspell the abbreviation for superoxide anion radical

In line 129, shouldn't it be respectively instead of separately?

In line 438 the authors write that visualisation of superoxide anion radical is with DHE but in fig 3 visualisation is with NBT - inconsistency

In addition, for me, it is not apparent that the blue colour at OE is weaker than at CK - fig 3

line 156 - I don't think this description is about the ROS content

Fig 5 - the photos do not clearly show the yellowing

Fig 9 - shouldn't there be content instead of abundance in chart A?

Fig 9 - what items C, D, G, H represent - how the visualisation was done - very unclear and confusing

line 271 - the authors do not specify the DHA content

Why are the activities of the same enzymes in fig 4 and fig 10 expressed in different units?

The description of the GSH results from Figure 12 is missing

Why are the contents of substances in boxes A and C as well as B and D in different units in Figure 12?

In my opinion, the work requires significant improvements and cannot be published in this form.

Round 2

Reviewer 2 Report

The authors referred to the list of comments I gave, but they did not refer to the text of my review. And the most important ones are :

However, throughout the paper I found no confirmation that the transformation was successful - and if indeed the authors were working with a plant overexpressing the gene CsASMT. I consider this to be a major shortcoming in the work and the lack of confirmation of a successful transformation prevents confidence in the results and puts into question the results obtained. In many places the style of language is strange, making it difficult to understand, in my opinion, the work needs language improvements. The authors do not state which statistical tests they use to assess the significance of the differences and whether the assumptions of these tests are met; they only state the software they use.

If the above points are not answered, the paper is not suitable for publication.

Round 3

Reviewer 2 Report

Dear Authors,

Now, I understand why information confirming transient transformation was not included. However, please add the information that such a confirmation has been made and presented in another publication.

In my opinion, after the additions, the work is suitable for publication.